# Decreasing the Likelihood of Multiple Organ Dysfunction Syndrome in Burn Injury with Early Antioxidant Treatment

**DOI:** 10.3390/antiox10081192

**Published:** 2021-07-26

**Authors:** Jackson P. Sterling, Vincent C. Lombardi

**Affiliations:** 1Department of Biology, University of Nevada, Reno, Reno, NV 89557, USA; jacksonsterling@nevada.unr.edu; 2Department of Microbiology and Immunology, University of Nevada, Reno, School of Medicine, Reno, NV 89557, USA

**Keywords:** vitamin C, vitamin E, NADPH, oxidase, MODS, ROS, histamine, xanthine oxidase, free radical

## Abstract

Major burn trauma initiates a cascade of physiological events that cause profound stress on the body, resulting in significant complications which often lead to death. An understanding of these events may afford earlier and more precise interventions which, in turn, may reduce these complications, thus, improving patient outcomes. Burn trauma is associated with numerous inflammatory events that result in the release of free radicals, which promote oxidative stress and subsequent tissue damage. These mass-inflammatory events affect the body systemically, leading to several detrimental responses including complement activation, excessive histamine release, decrease in blood pressure, release of reactive oxygen species, and ultimately multiple organ dysfunction syndrome (MODS). However, recent studies conducted on the use of antioxidants as a part of a burn treatment protocol have shown promising results. In this review, we will discuss the current research and advancements in the treatment of burn trauma with the use of antioxidants, and how the early administration of antioxidant can possibly reduce the risk of developing MODS.

## 1. Introduction

Every year, burn trauma accounts for over 300,000 deaths globally [1]. The annual prevalence requiring medical attention in the United States is estimated at 1.1 million cases; of these, approximately 40,000 require hospitalization with 20,000 being classified as major burns [2]. Major burns, which are defined as a burn that covers at least 20% of the patient’s total body surface area (TBSA), are typically classified as either first, second, third or fourth degree [3]. First degree burns are superficial, affecting the epidermis and do not typically require medical treatment. Second degree burns, which are more serious, are classified as superficial partial-thickness or deep partial thickness, relative to the depth of dermal layer damage. Third- and fourth-degree burns are characterized by the destruction of the entire thickness of the dermal layer and are classified as full-thickness [4,5]. It is well established that large burn injuries, those covering ≥20% of the TBSA, induce a host of major stressors and mass inflammatory events, including fluid loss, oxidative stress, hypermetabolism, immunosuppression, and organ dysfunction [6]. Indeed, most major burns meet the criteria for systemic inflammatory response syndrome (SIRS) [3]. The first indication of sepsis is in fact SIRS, and is defined as having two or more of the following symptoms: temperature above 38 °C or below 36 °C; a heart rate of at least 90 beats per minute (tachycardia); a breathing rate of at least 20 breaths per minute (tachypnea); a white blood cell count (WBC) greater than 12,000 per mm^3^ (leukocytosis); or less than 4000 per mm^3^ (leukopenia) or a change in bands (immature WBC) of at least 10% [3,7].

Given that burns are unique in comparison to other injuries, burn physicians have begun looking toward other, more precise signs for detecting sepsis such as renal dysfunction, increased fluid requirement, decreased platelet count, change in mental status, and declining pulmonary function [3]. It is common for a burn patient to quickly develop septic shock which is defined as sepsis with MODS [3]. In fact, MODS is the main cause of mortality in major burn cases, provided that the patient survives the first 24 h post-injury [3]; a period known as the early resuscitation phase [8]. During this time, the primary focus of medical intervention is to establish hemodynamic stability of the patient, which is achieved by fluid resuscitation to combat hypoperfusion of organ and muscle tissues; a serious consequence of the profound hypovolemia observed in the initial 24 h post-burn injury [3]. While fluid resuscitation is crucial for patient survival, it is common for physicians to administer excessive fluids, despite there being standardized protocols to calculate proper fluid amounts, such as the Parkland formula [3]. Over resuscitation can lead to abdominal compartment syndrome, as well as MODS [9,10]. Nevertheless, perhaps the most common cause of MODS associated with burn injury is the over-abundance of reactive oxygen species (ROS) produced by the systemic inflammatory events that occur following burn trauma [11,12]. This production comes from different sources, but all contribute to systemic organ failure. It is important to note, however, that while ROS can lead to major tissue damage, there are benefits, such as their antibacterial properties, that should not be ignored. Given the excessive production of ROS associated with major burn trauma, the cons do outweigh the pros. However, trying to mitigate the adverse consequences of these ROS while reaping their benefits is ideal. Recently, researchers have been conducting studies on antioxidant treatment for major burn trauma. Although there is not yet a broadly accepted standard protocol for such a treatment, these studies have shown promising results which may ultimately lead to such a standard protocol being developed in the future. The goal of this review is to encourage the establishment of such protocols by discussing the potential efficacy of antioxidants for the treatment of burn trauma.

## 2. Pathophysiology of ROS Production

Major burn trauma initiates a plethora of extraordinary events that shift the physiologic function toward an altered state as the body’s systems attempt to return to homeostasis. Of these, we will focus on the oxidative stress caused by the production and release of ROS, as well as the benefits and consequences which accompany them. While most ROS, in a healthy individual, are generated as byproducts of cellular metabolism, through the electron transport chain in mitochondria, or as necessary intermediates of metal-catalyzed oxidation reactions, ROS production in burn trauma stems from multiple, distinct sources. These include complement activation, the activation of intravascular neutrophils, and the increase in xanthine oxidase activity, which is upregulated through the mobilization of histamine by mast cells [13,14]. Although distinct, these sources are also interconnected to a large extent, and together contribute to the total ROS generated. 

Complement system activation, which is commonly associated with burn trauma [14], is an innate immune response that uses an amplifying cascade of signaling proteins and enzymes to promote inflammation as well as to opsonize pathogens and infected cells targeted for removal via phagocytosis. The most notable proinflammatory complement proteins relevant to burn trauma are the anaphylatoxins, C3a and C5a. Both proteins activate chemotaxis and granule enzyme release with respect to several cell types including macrophages, monocytes, neutrophils, and eosinophils; however, their activation of neutrophils appears to be responsible for the largest direct production of superoxide anions [15,16,17,18,19,20]. With respect to macrophages, neutrophils, and eosinophils, C3a and C5a can trigger a rapid release of the reactive oxygen species superoxide anions and hydrogen peroxide; this is commonly referred to as a respiratory burst, or an oxidative burst [14,21,22,23]. The respiratory burst typically occurs within the phagosome of macrophages and neutrophils upon the ingestion of a bacterium or on the surface of parasites, such as helminths, through the direction of eosinophils. Three receptors are known to be involved in C3a and C5a signaling: C3aR, C5aR1and C5aR2. Although C3aR and C5aR1 are both G-protein coupled seven transmembrane receptors, C5aR2 is not, and its function is not well characterized but studies in mice suggest it may be involved in an anti-inflammatory capacity [24]. Engagement of C3aR or C5aR1 by their cognate ligands, leads to chemotaxis, granule enzyme release, and bacterial opsonization. The activation of these polymorphonuclear granulocytes by C3a or C5a promotes the assembly of the multicomponent enzyme complex, NADPH oxidase. Granulocyte NADPH oxidase is composed of six subunits; two membrane-associated subunits, gp91^phox^ and p22^phox^, form the catalytic core whereas four subunits, p67, p47 and p40 and the Rho GTPase, typically either Rac1 or Rac2, are present in cytosol (Figure 1A). Upon activation, this oxidase catalyzes the production of a superoxide free radical by transferring one electron to oxygen from NADPH, during which, molecular oxygen is transported from the extracellular space to the cell’s interior and a proton is generated [21], (Figure 1B). 

In addition to their role in activating NADPH oxidase, C3a and C5a are potent inflammatory mediators that promote the activation of the NLRP3 inflammasome, as well as stimulating the release of histamine from mast cells [13,25]. Importantly, the triggering of mast cells by C3a and C5a to release histamine may be their most important contribution to the pathology of major burn trauma [16,26]. The release of histamine from mast cells leads to vascular permeability as well as the dilation of blood vessels [27] and once the capillaries become leaky, plasma seeps out of the vessels into the interstitial space, decreasing blood volume (hypovolemia). The combination of dilated blood vessels and reduced intravascular volume can lead to a significant and life-threatening reduction in blood pressure. The combined effects of these factors are the primary reason burn patients require copious amounts of fluid within the early resuscitation phase. The release of histamine has also been shown to promote ROS generation through reverse substrate-inhibition of xanthine oxidase [28]. This was initially shown by Friedel et al. using a rat model of thermal injury, whereby they reported that increased histamine corresponded with increased plasma xanthine oxidase as well as elevated xanthine oxidase in pulmonary artery endothelial cells [13]. However, it should be noted that unlike rats, humans do not display such high concentrations of xanthine oxidase in their plasma. Alternatively, in humans, histamine may interact with cellular sources of xanthine oxidase to promote increased xanthine oxidase activity [13]. 

## 3. Importance of ROS

The mass production of ROS, and subsequent oxidative stress, can have several detrimental effects on the body. ROS attack organ tissues, oxidize lipids, and impair cellular function, ultimately leading to MODS. Elevated ROS also leads to the production of inflammatory cytokines through several different mechanisms. For instance, activation of the NLRP3 inflammasome requires two signals, the first of which is a priming signal. This priming signal can occur through the engagement of pattern recognition receptors, (i.e., toll-like receptors), or by pathogen- or damage-associated molecular patterns (PAMP/DAMP). Inflammatory cytokines including TNF-α and IL-1β, which drive NF-κB nuclear translocation and subsequent transcription of the NLRP3 and IL1B genes, can also serve as the first signal. The second signal, which promotes the assembly of the inflammasome complex, is triggered by several different activators including calcium influx, uric acid crystals, extracellular ATP and ROS [29,30,31,32]. Assembly of the inflammasome leads to the activation of cellular caspase-1, which in turn cleaves pro-peptides generating the mature inflammatory cytokines IL-1β and IL-18 (reviewed in [33]). 

It is worth noting, however, that ROS can also be beneficial to burn patients when present in small amounts [34]. Research has shown that ROS kills microbes by attracting lymphocytes and helps promote wound healing by regulating angiogenesis [34,35,36,37,38], which is crucial for the wound healing process as it allows for more nutrients, especially oxygen, to enter the wound site. Increasing oxygen concentration to the damaged tissues is also crucial given that these tissues may become anaerobic and colonized by anaerobic bacteria [7,39]. Because there are several benefits, as well as injurious consequences to ROS production, there may be an ideal concentration that maximizes the positive effects while reducing the negative ones; albeit this theoretical ideal concentration of ROS in wound healing has yet to be defined [36]. Patients with major burn trauma are very susceptible to infection due to the loss of epidermal barrier integrity. Indeed, given the TBSA of the epidermis as well as its importance as a physical barrier against pathogens, minimizing infection while promoting wound healing is crucial when treating major burn traumas. Encouragingly, modern wound care protocols have been shown to significantly reduce the burden on the patient’s immune system, with respect to controlling infections. Since the 1950’s, mortality in major burn trauma has decreased by over 50% due to such advancements in wound care [40]. While it could be argued that the advancements in wound care and sanitary protocols have decreased the significance of ROS in infection control, its contributions to angiogenesis, as well as inflammation, remains to be a critical issue. Accordingly, determining the ideal amount of ROS for wound healing in burns is a multifactorial question and further research is needed. 

## 4. Burn Treatment with Antioxidants

### 4.1. Endogenous Antioxidants

Several endogenous non-enzymatic antioxidants contribute to the elimination or limitation of ROS. For instance, through its ability to act as a scavenger of oxidant and electrophilic species, either directly or through enzymatic catalysis, glutathione may provide the most significant intracellular protection against the harmful effects of ROS. Although glutathione is oxidized during its reactions with ROS, it is reduced back to its biologically active form through a redox reaction involving NADPH-dependent reductase [41]. Uric acid, is a hydrophilic antioxidant produced during purine metabolism, and accounts for approximately two thirds of the total oxygen scavenging activity in the blood [42]. Nevertheless, previous studies have also suggested that hyperuricaemia, a situation associated with burn trauma, may have a detrimental pro-oxidant effect [43]. Therefore, the contribution of uric acid in promoting or inhibiting oxidative stress in burn trauma may not be straight-forward and further studies are needed to evaluate its contribution. Several other non-enzymatic endogenous antioxidants contribute to limiting ROS, many of which are found in blood plasma. They include proteins such as ferritin, transferrin, ceruloplasmin, and albumin as well as several metallothioneins. Metallothioneins are a group of small conserved, metal-binding proteins, rich in the amino acid cysteine, which allow them to bind several trace metals, such as copper, iron, and zinc via their -SH functionality (Reviewed in [44]). In addition to their role in protecting against oxidative stress, metallothioneins also provide an important buffering function against toxic heavy metals [45]. Together, these molecules provide an important first line of defense in limiting the formation of ROS.

In addition to non-enzymatic antioxidants, a host of enzymatic antioxidants play important roles in the elimination of ROS. The superoxide dismutases (SODs) catalyzes the dismutation of the superoxide radical into molecular oxygen and hydrogen peroxide. The importance of these enzymes in cellular protection can be emphasized by the fact that they are the most catalytically efficient enzymes known, having the largest kcat/KM (an approximation of catalytic efficiency) observed for any enzyme (~7 × 10^9^ M^−1^s^−1^) [46]. In fact, the dismutation of the superoxide radical is only limited by the frequency of collision between itself and SOD. Given that superoxide is one of the principal reactive oxygen species in major burn pathology, it is not surprising that exogenously administered SOD has been studied in animal models of thermal injury. Indeed, Sáez and coworkers reported that prior administration of superoxide dismutase or the xanthine oxidase inhibitor allopurinol increased the survival rates of mice subjected to burn shock; however, the same effect was not observed when the treatment was administered after thermal injury. This observation raises the possibility that the effect was not due to the reduction in reactive oxygen species, but perhaps through prophylactically preventing elevated blood uric acid production, and, in turn, preventing acute kidney injury that results from elevated blood uric acid. Indeed, acute kidney injury is a common comorbidity in severely burned patients. Previous studies suggest that acute kidney injury associated with burn trauma occurs in approximately 30% of such patients admitted to intensive care units and is commonly associated with a mortality as high as 80% [47,48,49,50]. Other important enzymatic antioxidants include glutathione peroxidase, glutathione reductase, and catalase, all of which neutralize hydrogen peroxide, into water and molecular oxygen molecule.

### 4.2. Dietary Antioxidants and Natural Products

Many biologically important antioxidants are exogenous and are obtained through the diet, such as flavonoids and phenolics, which are readily available in fruits and vegetables [51]. However, the two most common exogenous antioxidants are probably vitamin C, (also known as ascorbic acid) and vitamin E (α-tocopherol). Vitamin C is a hydrophilic molecule and primarily acts interstitially while vitamin E, being lipid soluble, plays a critical role in protecting intracellular membranes and organelles. In addition to its function as an antioxidant, vitamin C plays a prominent role in several metabolic processes, such as the activation of folic acid, and the conversion of the amino acid, tryptophan, to the neurotransmitter, serotonin [52]. Likewise, vitamin E also has many metabolic functions in addition to its important antioxidant role, including the reduction prostaglandins such as thromboxane, which promote platelet clumping [53]. 

Astaxanthin (3,3’-dihydroxy-b, b’-carotene-4,4’-dione) is a natural keto-carotenoid with broad antioxidative properties. It is produced by microalgae and accumulates in zooplankton, which feed on these algae, as well as several species of fish which consume the zooplankton. It is also responsible for the characteristic pinkish color of shrimp, Pacific salmon, and trout. Using a rat model of thermal injury, Guo and coworkers reported that the administration of astaxanthin attenuates early acute kidney injury following severe burns by mitigating oxidative stress and mitochondrial-related apoptosis [54]. In a subsequent study, they observed that astaxanthin imparts protection from early acute kidney injury in severely burned rats by inactivating the TLR4/MyD88/NF-κB axis and upregulating heme oxygenase-1.

Although the body contains its own supply of antioxidants, such as glutathione, in major burn patients, the body’s supply is quickly depleted in response to the sudden spike in ROS production. For this reason, the elevated concentrations of ROS promote systemic oxidative stress, affecting most organs, ultimately leading to MODS [6]. Due to the high production of ROS in such circumstances, administration of antioxidants would seem to be a logical course of action. Indeed, antioxidants would neutralize excess ROS, and potentially reduce or even possibly prevent some of the deleterious effects associated with major burn trauma. Multiple studies conducted on the effects of exogenous antioxidants in burn patients confirm this supposition and have shown promising results, such as improved wound healing, decrease in organ dysfunction, and a decrease in mortality [55]. In a study assessing the benefits of exogenously administered glutathione, Buz et al. treated Wistar rats mesotherapeutically (injected into the subcutaneous fat) with glutathione, as well as taurine, and L-carnitine and observed that glutathione mesotherapy was the most effective when used to treat partial-thickness thermal burns; however, all treatments were statistically significant when compared to the sham treatment (saline) and the non-treated control group (*p* = 0.001).

Rehou et al. reported on the beneficial effects of administrating an intravenous cocktail consisting of multivitamins and trace elements such as selenium and zinc in Ringer’s lactate [6]. In short, the treatment was administered for the first 14 days postburn, and then two multivitamins per day as well as two 200 mcg doses of selenium beginning on day 15 onward. While the researchers did not investigate specific changes in morbidity or mortality, a significant average decrease in the length of the patient’s hospital stay was observed; thus, suggesting the treatment was clinically efficacious. Likewise, Al-Jawad and coworkers reported that antioxidant supplementation promoted wound healing as well as decreased overall mortality [55]. Specifically, blood samples were collected from each patient within the first 24 h of admission, and before the administration of any antioxidants. Six groups of 30 subjects were treated with antioxidants (vitamin C, vitamin E, zinc sulphate, allopurinol, melatonin, or N-acetylcysteine). Additionally, two control groups were included: one treated according to standard medical practices and one non-burn control. A significant improvement was observed in the referenced parameters when compared to the control groups; with the greatest benefit observed for allopurinol, melatonin, and N-acetylcysteine, with respect to decreased mortality rate and melatonin and N-acetylcysteine, with respect to healing time. However, all antioxidant treatments showed significant benefit when compared to the non-treated controls. While this study did have an improved patient outcome when compared to the Rehou et al. study, there are important differences. For example, the study of Al-Jawad et al. included patients with a TBSA range from 15–40% burned without specifications on burn depth, while Rehou et al. investigated patients with ≥ 30% burn, ≥ 20% full thickness burn, or ≥ 20% with required mechanical ventilation and a total average TBSA of 33 +/− 13% [6]. It is possible that the improved outcomes reported by Al-Jawad et al. were due to either more effective antioxidant treatments or having used less severe burn cases. Albeit, the use of zinc, N-acetylcysteine, and both vitamins C and E were common in both studies. 

While many different antioxidants have the potential to be efficacious in treating burn patients, vitamin C has numerous advantages for use in the clinical setting, given its low toxicity, its low cost and its commercial availably. Importantly, vitamin C is water-soluble, so excess is excreted by the kidneys, preventing toxic buildup. Although vitamin C is water soluble and cannot scavenge ROS within the cell membrane, it can regenerate vitamin E, which is lipid soluble and has the capacity to scavenge ROS in the cell membrane. Indeed, both vitamins C and E have been shown to neutralize free radicals in burn patients. Vitamin C primarily does so in the interstitial space while vitamin E does so within the cells [56,57]. In contrast to vitamin C, which has direct antioxidant properties, N-acetylcysteine has been shown to increase the levels of endogenous antioxidants such as glutathione while decreasing inflammatory mediators such as TNF-α, IL-1β, and CXCL8 (IL-8) [56,58]. Likewise, previous studies have reported that zinc can improve the body’s natural supply of an antioxidant glutathione, which improves both the mortality rate as well as wound healing time [59]. Improvement of wound healing time as well as a lower mortality rate would greatly improve the treatment and outcome of burn patients.

Given that there is a high risk of infection for severe burn cases (provided that the patient survives the first 24 h post-injury), understanding the role of antioxidants in preventing infection represents an important point of consideration. In a follow-up study to that of Al-Jawad et al., the same authors, using the same patient cohort, reported that, in addition to the decreased mortality rate, antioxidant treatment was associated with reduced incidence of wound infection and improved healing time [60]. It was observed that patients who received a daily dose of either 400 mg of vitamin E with 500 mg of vitamin C, or 75 mg of zinc sulfate, or 100 mg of allopurinol or 500 mg of N-acetylcystine showed a significant decrease in invading microorganisms of the patient’s wounds, when compared to non-treated controls. However, it should be acknowledged that the observed results may not be directly related to their antioxidant function; the effect may be indirect through the promotion of non-antioxidant pathway. For instance, in addition to being an antioxidant, vitamin C is an essential cofactor for collagen biosynthesis, as well as carnitine and catecholamine metabolism, and it also enhances the absorption of dietary iron [61]. Improved collagen biosynthesis would directly restore the protective epithelial barrier [62,63] and improved carnitine and catecholamine metabolism may benefit the burn patient by impacting core temperature [64]. Additionally, improved iron adsorption may increase hemoglobin production and reduce anemia associated with burn trauma [65]. Therefore, while these results are highly encouraging, more specific studies are warranted to definitively resolve the underlying mechanisms. A summary of several discussed antioxidants and the ROS which they neutralize are given in Table 1.

## 5. Conclusions

Recent research conducted on antioxidant supplementation in major burn trauma has been shown to improve certain aspects of patient recovery such as an average decrease in length of the hospital stay, increased wound healing rates, and an overall decrease in mortality. It is well known that the high concentrations of ROS present in burn trauma patients commonly leads to MODS and death. However, the benefits of having small amounts of ROS present for the wound healing process should not be neglected. An ideal concentration of ROS would maximize the wound healing benefits while minimizing the adverse consequences; however, such knowledge is currently lacking. Moreover, previous studies indicate that the administration of antioxidants as soon as possible is beneficial to combat ROS production; however, the optimal concentrations as well as the optimal combination of antioxidants have not been rigorously established. Therefore, further studies to optimize the treatment protocols are needed. 

Since the major cause of mortality of major burn victims that survive the early resuscitation phase is MODS, larger studies may provide more evidence regarding how antioxidants may be used to reduce the likelihood of developing this comorbidity. Based on the consequences and pathophysiology of ROS production and the amount produced in major burn trauma, it is reasonable to hypothesize that antioxidant treatment may significantly reduce these negative effects. Due to the high concentrations of free radicals, natural antioxidant supplies are quickly depleted, and supplementation could prove to be beneficial. Moreover, it has been shown that patients who suffer from burn trauma produce less endogenous antioxidants; therefore, increasing their endogenous production may also represent an alternative or supplemental strategy to burn treatment. Given that there are roughly 20,000 major burn cases in the US annually, improved treatment protocols are needed and warranted. Many antioxidants, such as intravenous vitamin C, are cheap, well-tolerated and are easily administered. Although the studies articulated in this review support the potential benefits of antioxidants in treating burn patients, standardized protocols that can be rigorously followed by emergency physicians are still needed. Indeed, the use of antioxidants to treat burn patients may significantly improve patient outcomes, and their use warrants further study and clinical validation.

## Figures and Tables

**Figure 1 antioxidants-10-01192-f001:**
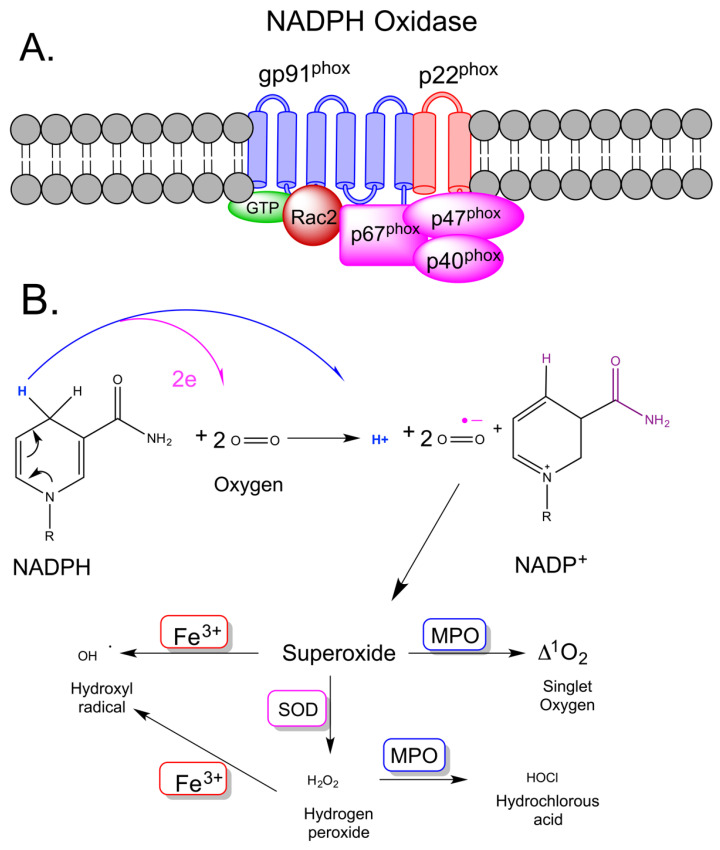
NADPH Oxidase. In resting neutrophils, the cytochrome b558 subunits (gp91^phox^ and p22^phox^) of the NADPH oxidase are localized in the plasma membrane. The other components of the oxidase (p40 ^phox^, p47 ^phox^, and p67 ^phox^) are located in the cytosol. Engagement of the complement component C5a receptor 1 by the compliment protein C5a (not shown), activates Rac2 and induces the assembly of the complete, active NADPH oxidase (**A**). Upon activation, NADPH oxidase transfers an electron from NADPH, via its FAD cofactor, to molecular oxygen, generating a superoxide radical ^•^O_2_^−^ ((**B**), top). Granule-localized ^•^O_2_^–^ is converted by superoxide dismutase (SOD) to microbicidal hydrogen peroxide (H_2_O_2_), which, in turn, can be converted by myeloperoxidase (MPO), a heme-containing enzyme, to microbicidal hypochlorite (OCl^–^) and by chemical reaction with ferrous (Fe^2+^) ions to the hydroxyl radical (^•^OH). MPO can also convert ^•^O_2_^–^ into singlet oxygen ((**B**), bottom).

**Table 1 antioxidants-10-01192-t001:** Antioxidants and the reactive oxygen species which they neutralize.

	Hydroxyl Radical	Superoxide Radical	Hydrogen Peroxide	Lipid Peroxides
Endogenous non enzymatic	Uric acid [42]	✓	✓		
Transferrin [66]	✓	✓	✓	
Albumin [67]	✓	✓	✓	
Ferritin [66]	✓	✓	✓	
Glutathione [68]			✓	
Ceruloplasmin [69,70]	✓	✓	✓	✓
Lipoic acid [71]	✓	✓	✓	✓
Bilirubin [72]	✓	✓	✓	
Melatonin [73]	✓	✓	✓	✓
Endogenous enzymatic	Superoxide [74] dismutase		✓		
Glutathione [75] peroxidase			✓	✓
Catalase [74]			✓	
Dietary	Vitamin C [76]	✓	✓	✓	
Vitamin E [76]			✓	✓
Vitamin A [77]	✓	✓	✓	✓
Beta-carotene [77]			✓	✓
Polyphenolics [78]	✓	✓		
Flavonoids [78]	✓	✓

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
