# Peer review of "Decreasing the Likelihood of Multiple Organ Dysfunction Syndrome in Burn Injury with Early Antioxidant Treatment"

_antioxidants, 2021, doi:10.3390/antiox10081192_

Round 1
Reviewer 1 Report
The authors focalize their attention on the burn trauma and offer a very interesting review. The review in general offer the possibility to the reader to learn many things about this problem. A detailed chapter is dedicated to physiological events of burn trauma that cause profound stress on the body, resulting in significant complications which often lead to death. The authors reflect also on the use of antioxidants as a part of a burn treatment protocol that have shown promising results.
Author Response
We wish to thank the reviewers for they helpful and insightful comments and suggestions. Each one was carefully considered and addressed, which in our opinion, significantly improve the manuscript. We hope that these changes as articulated below, meet with the approval of the reviewers.
Kind regards,
Vincent C. Lombardi
In response to Reviewer 1, we have carefully proofed the manuscript and made the suggested grammatical changes. These changes are highlighted in red.
Reviewer 2 Report
The review explores the role of anti-oxidant treatment.
The authors need to rephrase some parts of the ms. In the introduction, the authors need to state clearly the goals of the ms. Conclusion section could be more exaustive.
Moreover, the authors need to explore other natural compounds that can work as anti-oxidants.
Author Response
We wish to thank the reviewers for they helpful and insightful comments and suggestions. Each one was carefully considered and addressed, which in our opinion, significantly improve the manuscript. We hope that these changes as articulated below, meet with the approval of the reviewers.
Kind regards,
Vincent C. Lombardi
Reviewer 3 Report
The study, titled “Decreasing the Likelihood of Multiple Organ Dysfunction Syndrome in Burn Injury with Early Antioxidant Treatment.” This review discusses the promising results of using antioxidant treatment for burn injury to reduce the risk of multiple organ dysfunction syndrome. The paper in its current form is unsuitable for publication; significant changes necessitate rewriting the paper.
Comments:
- I find the organization with respect to topic or theme or outline to be rather poor. There is no overarching hierarchy (or, if there, it is not obvious). The manuscript would benefit from an outline in the introduction so that a reader could find their way to the subtopic of particular interest.
- The title "Antioxidant Treatment and Its Effects on Burn Patients." Please discuss the Importance of Antioxidant Treatment.
- Figure 1 is not explained very well.
- Authors must use a flow chart to explain each title discussion section. The reader will find it simple to read and understand.
- The author intends to describe the limitations of the other treatment methods in the introduction section.
- Finally, the dosage, timing, and duration period for antioxidant treatment remain unknown. The author wishes to discuss "Antioxidant Treatment and Its Effects on Burn Patients" in the discussion section.
Author Response
We wish to thank the reviewers for they helpful and insightful comments and suggestions. Each one was carefully considered and addressed, which in our opinion, significantly improve the manuscript. We hope that these changes as articulated below, meet with the approval of the reviewers.
Kind regards,
Vincent C. Lombardi
In response to Reviewer 3, we have reorganized the layout of the article as suggested; we have expanded our discussion regarding the importance of antioxidant treatment; we have improved the explanation and ledged of Figure 1; we have added a discussion regarding the limitations of other methods and we discussed the idea of dosages, timing, and duration of antioxidant treatment, while making an effort not to give specifical medical advice, which is beyond the scope of the article. These changes are highlighted in red.
Round 2
Reviewer 2 Report
I suggest to accept the ms
Author Response
Thank you!
Reviewer 3 Report
This review discusses the promising results of using antioxidant treatment for burn injury to reduce multiple organ dysfunction syndrome risk. The study, titled “Decreasing the Likelihood of Multiple Organ Dysfunction Syndrome in Burn Injury with Early antioxidant Treatment.” I recommend publication of this review in antioxidants after the minor revision.
Comments:
- The authors did not include a flow chart for each subtitle of Burn Treatment with Antioxidants in the revised manuscript in response to a comment (4).
- In addition, in response to a comment (6), the authors should discuss the dosage, timing, and duration of antioxidant treatment in the revised manuscript.
Author Response
Dear reviewer,
Thank you for your comments. The modified manuscript shows the requested changes in red. We have added an additional subheading titled "Dosage, timing, and duration of treatment." We have added specific details on which antioxidants was used, the dosages and the duration of treatment. We have also identified and referenced a specific protocol for their use, however, in order not to violate copyright we only given a brief overview of this protocol.
Lastly, with respect to the reviewers request to include a flow chart for each subtitle, I think I'm a bit unclear on exactly what you would like to see. If the reviewer could provide an example of this in another publication, I would be happy to attempt to emulate this and add it to our manuscript.
Thank you again for your help.